# The State of Art of Regenerative Therapy in Cardiovascular Ischemic Disease: Biology, Signaling Pathways, and Epigenetics of Endothelial Progenitor Cells

**DOI:** 10.3390/cells9081886

**Published:** 2020-08-11

**Authors:** Fabio Perrotta, Angelica Perna, Klara Komici, Ersilia Nigro, Mariano Mollica, Vito D’Agnano, Antonio De Luca, Germano Guerra

**Affiliations:** 1Dipartimento di Medicina e Scienze della Salute “V.Tiberio”, Università del Molise, 86100 Campobasso, Italy; angelica.perna@unimol.it (A.P.); klara.komici@unimol.it (K.K.); germano.guerra@unimol.it (G.G.); 2Dipartimento di Scienze e Tecnologie Ambientali, Biologiche, Farmaceutiche, Università della Campania “Luigi Vanvitelli”, 81100 Caserta, Italy; nigro@ceinge.unina.it; 3CEINGE-Biotecnologie avanzate, 80145 Naples, Italy; 4Dipartimento di Scienze Mediche Traslazionali, Università della Campania “Luigi Vanvitelli”, 80131 Naples, Italy; mollicamariano@gmail.com (M.M.); v.dagnano@studenti.unimol.it (V.D.); 5Department of Mental and Physical Health and Preventive Medicine, Section of Human Anatomy, University of Campania “Luigi Vanvitelli”, 80138 Naples, Italy; antonio.deluca@unicampania.it

**Keywords:** EPCs, cardiovascular ischemic disease, regenerative medicine, calcium homeostasis, regenerative therapy, epigenetic

## Abstract

Ischemic heart disease is currently a major cause of mortality and morbidity worldwide. Nevertheless, the actual therapeutic scenario does not target myocardial cell regeneration and consequently, the progression toward the late stage of chronic heart failure is common. Endothelial progenitor cells (EPCs) are bone marrow-derived stem cells that contribute to the homeostasis of the endothelial wall in acute and chronic ischemic disease. Calcium modulation and other molecular pathways (NOTCH, VEGFR, and CXCR4) contribute to EPC proliferation and differentiation. The present review provides a summary of EPC biology with a particular focus on the regulatory pathways of EPCs and describes promising applications for cardiovascular cell therapy.

## 1. Endothelial Progenitor Cells (EPCs): Definition 

Stem cells are defined by both self-renewal capacity and the ability to differentiate into mature tissue-specific cells [1]. Endothelial progenitor cells (EPCs) refer to populations of cells that are capable of differentiation into mature endothelial cells and vasculogenesis (de novo formation of vascular networks) [2]. There is, however, a controversy between this correct theoretical definition and the scientific evidence for the identification and definition of the role of these cells, causing numerous cell types to be named as EPCs [3]. The scientific studies, carried out by different research groups, could not be easily compared, leading to conflicting results and hindering scientific progress; the researchers considered EPCs to be very heterogeneous, creating confusion about the role of these cells [4]. EPCs were first identified and isolated from human peripheral blood by Asahara and colleagues [5]. They used the terminology “putative EPC”; therefore, different names to describe the endothelial progenitors entered the use of the scientific literature, causing a certain degree of confusion; to overcome this, it is important to know that there are two approaches used to study endothelial progenitor cells, one based on the use of flow cytometry on blood samples and the other one on isolation methods of cultured cells [6]. In flow cytometry, circulating EPCs are identified as mononuclear cells expressing CD34, vascular endothelial growth factor receptor 2 (VEGFR2), and CD133 [7,8], while in cell culture methodologies two different populations are distinguished, namely, hematopoietic and endothelial [9], using the terminologies “hematopoietic EPC” and “non-EPC hematopoietic”, respectively [10]. Hematopoietic EPCs include circulating angiogenic cells (CACs), pro-angiogenic hematopoietic cells (PACs), colony-forming unit-Hill (CFU-Hill) EPCs, early EPCs, early outgrowth EPCs, hematopoietic EPCs, small EPCs, myeloid EPCs, and myeloid angiogenic cells (MACs) grown under endothelial culture conditions; and non-hematopoietic EPCs include endothelial colony-forming cells (ECFCs), outgrowth endothelial cells (OECs), blood outgrowth endothelial cells (BOECs), endothelial outgrowth cells (EOCs), late EPCs, late outgrowth EPCs, non-hematopoietic EPCs, and large EPCs. In these studies, it is difficult to have significant comparisons, as the purity and the mechanisms of action of the cells differ considerably; different cell types can play different roles, and synergistic effects are observed when using distinct EPCs together. The studies could be made easier by grouping cells into two main categories: hematopoietic (myeloid) and endothelial [11]. To be able to have some clarity, it would be necessary to define a specific phenotype and a defined biological function; the term EPCs should be replaced by the term ECFCs [12,13], as it describes well the phenotype and function of these cell types. MACs and CACs are not endothelial or progenitor cells, but are myeloid cells with pro-angiogenic and vasoreparative function, through a paracrine mechanism. MACs do not give rise to endothelial cells, but remain true to their hematopoietic nature [14,15]. Techniques should be standardized to evaluate the ability to form a vascular network in vitro and in vivo and a detailed identity immunophenotype (positive for CD31, VE-cadherin, von Willebrand factor, CD146, and VEGFR2; and above all, negative for CD45 and CD14) for cell characterization. Furthermore, clonogenicity and proliferative capacity should become standard criteria for distinguishing true progenitors from mature endothelial cells [16,17]. It is important to determine whether there is an organ specificity for the progenitors and to determine the niche in which the progenitors reside.

## 2. Origin and Biological Significance of EPCs

Over the past two decades, it has become clear that EPCs comprise a mixed population of heterogeneous cells in terms of lineage, proliferative potential, and mechanism of action; and an isolation method through the expression of surface markers has not yet been established, as a single panel of markers is still being studied. Currently, the common isolation method of EPCs is represented by the seeding of mononuclear cells (MNCs) on a plate covered with fibronectin and/or gelatin with medium containing vascular endothelial growth factor (VEGF). In this approach, two different types of cells have been recognized and are described according to their time of appearance in culture, morphologies, and expression of proteins: the “spindle-shaped” ECs, also called early EPCs (eEPCs), that appear in culture after 1–2 weeks and which share more monocyte surface antigens (CD45, CD14, and CD31), while they are negative for CD133, CD146, and Tie2 (Caplan, 2017), and the “cobblestone” morphological cells, also called late EPCs (lEPCs) or endothelial colony-forming cells (ECFCs) [16,18], which lack hematopoietic and myeloid markers and usually result from long-term cultures of at least 2–4 weeks in vitro [19].

The eEPCs are the “less potent” progenitor type since they are not able to differentiate into functional endothelial cells, but predominantly promote vessel formation by activating resident endothelial cells through paracrine mechanisms [20,21]. In normal cases, eEPCs are extremely low in blood. However, their concentration is influenced by various exogenous factors, endogenous factors, and pathological conditions. Indeed, their purpose is the restoration of endothelial function through paracrine means, as they lack direct vasculogenic effects [22]. ECFCs are “more potent” progenitor cells compared with eEPCs as they can generate mature endothelial progeny in vitro and contribute to the formation of new capillaries [23]. Similar to eEPCs, ECFCs are very low in number in blood, but they peak in numbers after an insult to repair vessels, thanks to their direct vasculogenic effects along with their production of angiogenic factors [21]. The origin of ECFCs is still controversial. They are likely mobilized bone marrow endothelial cells (BMECs), that originated as resident progenitor cells in a niche in the bone marrow (BM) rather than from tissue vascular niches [24]. The stimuli and mechanisms that drive the mobilization are not fully clear, although insults such as ischemia have been proven to massively induce the release of ECFCs into the bloodstream [25,26,27].

It is clear that the definition of early and late EPCs, based on their time of appearance in culture, reflects a very different phenotype, one being hematopoietic and the other endothelial, respectively [9]. It is also obvious that the starting definition of EPCs as circulating cells with the ability to differentiate into mature endothelial cells and contribute to endothelial repair at sites of vascular damage is not enough precise. Some of the populations originally defined as EPCs do not fulfill this definition, as the “early” EPCs promote angiogenesis through paracrine mechanisms, but cannot give rise to mature endothelial cells. While eEPCs and ECFCs are intrinsically different lineages of cells, they cooperate in the re-vascularization process. First, circulating eEPCs are delivered into the damaged tissue and their paracrine factors recruit ECFCs from either the circulation or the local vascular wall [10]. Next, the migration and proliferation of ECFCs are guided by MACs, and the ECFCs are recruited to the injured site to restore the endothelial integrity of the vascular wall [28].

From an applicative point of view, ECFCs are very attractive due to their robust proliferative and clonal ability, granting the generation of sufficient number of cells for clinical application in a relatively short period. ECFCs can exert their therapeutic potential through more than one molecular mechanism. The prevalent strategy is the direct physical incorporation into neovessels of the target tissue, thus improving vessel reconstruction, oxygenation, and delivery of nutrients [19]. ECFCs also act through a paracrine manner by creating a niche for the differentiation of stem/progenitor cells [29]. It is more likely that the two mechanisms cooperate in order that ECFCs can repair vasculature damages [30,31]. For the regenerative medicine applications, two major approaches have been used in vivo for the delivery of the cells, namely, cell bolus into the systemic circulation or targeted tissue, or cell-embedded biocompatible scaffolds. 

## 3. Signaling Pathways Driven by and Affecting EPCs

As above mentioned, in adults, ECFCs are thought to be quiescent for years, but capable of forming new blood vessels under ischemic and/or hypoxic conditions by sprouting new ECFCs from pre-existing vessels. Understanding mechanisms that drive the proliferation and differentiation of ECFCs will be useful in the context of regenerative medicine [26,32].

Moreover, the isolation of ECFCs is very difficult and patient-dependent and has a low yield. Currently, there are no markers that uniquely identify human ECFCs, making hard the isolation and cultivation of these cells. However, several panels of markers have been proposed to efficiently isolate ECFCs. Although firstly identified as CD34^+^, later on it has been demonstrated that the frequency of this marker is variable and ECFCs might lose the expression of this marker as the cells are expanded in vitro [33]. ECFCs are also characterized by the expression of endothelial markers such as CD31, CD146, VEGFR2, von Willebrand factor (vWF), kinase insert domain receptor (KDR), and VE-cadherin. Importantly, ECFCs are negative for CD14, CD45, CD115, and AC133 that are markers of hematopoietic cells, allowing a clear differentiation between the two cell types. It is important to take into account that the endothelial markers do not specifically mark ECFCs since CD34^+^/VEGFR^+^ may also identify circulating mature endothelial cells shed from damaged vessels. Subsequently, CD133 has been included as a stemness marker of ECFCs; however, the use of CD133 remains controversial [34]. Recently, tissue-resident vascular endothelial stem cells (VESCs), positive for CD157, have been identified [35]. The authors found that in response to liver vascular damage or as a part of the normal physiological turnover during liver homeostasis, CD157-positive VESCs undergo a proliferative expansion and sequentially regenerate vessels, reflecting their stem cell properties [35]. Further studies are needed to clarify the use of CD157 as a marker. Beyond the difficulty in isolating ECFCs from blood, it is also critical to improve their survival, homing ability, and ability to migrate, differentiate, and secrete pro-angiogenic factors in order to maximize their application in regenerative medicine [36,37]. To achieve the goal, researchers are exploiting the following routes: for example, several signal transduction pathways (e.g., C-X-C chemokine receptor type 4 (CXCR4), vascular endothelial growth factor receptor (VEGFR), sonic hedgehog (SHH), NOTCH, and wingless-type mouse mammary tumor virus integration site (WNT) seem to coordinate the survival, differentiation, and blood-vessel morphogenesis [38,39,40] (Figure 1). Modifying and modulating some of these signaling pathways may improve ECFC utilization in the regenerative medicine field. 

### 3.1. ECFCs and Calcium Homeostasis 

One of the most relevant signals that control ECFC proliferation and differentiation is the Ca^2+^ concentration. An increase in intracellular Ca^2+^ concentration has long been known to play a crucial role in angiogenesis [41,42,43,44]. Similar to mature endothelial cells, ECFCs require an increase in Ca^2+^ to proliferate, assemble into capillary-like tubular networks in vitro, and form neovessels in vivo [41]. There are three main Ca^2+^-transporting systems, the plasma membrane Ca^2+^ ATPase, the Na^+^/Ca^2+^ exchanger (NCX), and the sarco-endoplasmic reticulum Ca^2+^ ATPase (SERCA), which sequester cytosolic Ca^2+^ into the endoplasmic reticulum (ER). Due to their close proximity to the blood system, ECFCs are constantly in contact with and stimulated by soluble factors. Most of those factors act through regulation of Ca^2+^ concentration. For instance, epidermal growth factor (EGF) triggers pro-angiogenic oscillations in Ca^2+^ in rat coronary microvascular endothelial cells [45]. On the other hand, VEGF elicits a transient increase in Ca^2+^ [46].

### 3.2. NOTCH

The ECFC hierarchy is defined by Notch signaling that represents a key factor driving cell cycle, progenitor quiescence, and self-renewal potential. It has been demonstrated in murine models that NOTCH overexpression in stromal cells substantially ameliorates the repair potential of ECFCs [47,48]. However, Kim et al. underlined that Notch-stimulated preconditioning in vitro failed to enhance ECFC vasculogenesis in vivo. In contrast, in vivo co-implantation of ECFCs with stromal cells that constitutively expressed the Notch ligand resulted in ECFC-derived increased vessel density and enlarged vessel area in vivo [47]. On the other hand, blocking the NOTCH pathway resulted in a loss of self-renewal and high-proliferative potential (HPP) colony formation capacity, reflecting progenitor exhaustion [49]. 

### 3.3. VEGFR 

ECFCs derived from adult peripheral blood had enhanced sprouting angiogenic potential in vitro and in vivo through upregulation of the VEGFR2 signaling pathway. Indeed, VEGFR2 activation stimulated endothelial cell tubulogenesis [50]. This effect was mediated by Ca^2+^ oscillations as VEGF-induced ECFC proliferation and tubulogenesis were inhibited by the Ca^2+^-EDTA chelant [51].

### 3.4. WNT

WNT signaling is widely known to modulate the stem cell niche and stem cell proliferation, expansion, and differentiation [52,53,54]. Smadja et al. found that WNT antagonists enhance the proliferation of ECFCs and their capacity to differentiate [55]. These effects have been attributed to the enhancement of VEGFR2, stromal cell-derived factor 1 (SDF-1), and CXCR4.

### 3.5. CXCR4

SDF-1a/CXCR4 signaling is considered to play a central role in mobilizing endothelial progenitors from bone marrow [56]. Their production increases after an endothelial injury and can regulate endothelial progenitor homing. SDF-1a augments ECFC adhesion and migration to the bloodstream through the upregulation of E-selectin [57]; the latter mediates the adhesion and migration of ECFCs following endotoxic endothelial injury in a CXCR4-dependent process [57]. In addition, it has been suggested that CXCR4 expression levels in ECFCs could be a predictive marker for the success of ECFC-based angiogenic therapy. In fact, a previous research found that ECFCs isolated from different donors showed differences in CXCR4 expression that linearly correlated with SDF-1a-induced migratory capacity [58].

## 4. Other Strategies in ECFC Modulation: Mesenchymal Stem Cells (MSCs) and Epigenetic

Recent studies suggested a key role for mesenchymal cells and particularly for MSCs in ECFC regenerative potential [59]. Shafiee et al. demonstrated that ECFCs primed with MSCs have enhanced engraftment and capillary formation potential; this is through an improved resistance mechanism to stress conditions and transition to a more mesenchymal differentiated phenotype, in part related to activated NOTCH signaling [60]. However, the efficacy of MSCs seems to be donor-dependent and needs to be further demonstrated [61]. 

Several miRNAs in the last decade have been associated with ECFC potential in terms of regeneration. miRNA21 (miR-21), miRNA27a (miR-27a), miRNA27b (miR-27b), miRNA126 (miR-126), and miRNA130a (miR-130a) levels have been associated with poor or strong differentiation in vitro of ECFCs and therefore with a different regenerative potential [62,63]. DNA methylation and histone modifications are other epigenetic mechanisms possibly involved [64]. Therefore, epigenetic drugs may lead to the simultaneous activation of multiple pro-angiogenic signaling pathways (VEGFR, CXCR4, WNT, NOTCH, and SHH) that stimulate ECFCs, representing potential tools to improve ECFC utilization. This in turn results in improved capacity of ECFCs to form capillary-like networks in vitro and in vivo. Thus, ex vivo treatment with epigenetic drugs has been proved to increase the vascular repair properties of ECFCs through transient activation of pro-angiogenic signaling pathways [65].

## 5. Myocardial Revascularization in Animal Models Using EPCs 

EPCs are increasingly regarded as game changers in the therapy of cardiovascular diseases (CVDs), especially if they are merged with new and more innovative strategies. In the recent past, myocardial tissue engineering has been regarded as a potential turning point for the clinical history of patients with CVDs. During this time frame, cell sheet engineering came into play [66,67]. Dergilev et al. tested cardiac progenitor cell (CPC)-based cell sheet transplantation in improving heart function after acute myocardial infarction (AMI) in rats. They documented a significant decrease in scar size combined with a thickening of infarcted wall (*p* < 0.05) in the cell sheet group compared with controls [68]. Hamdi et al. compared intramyocardial injection of adipose-derived stromal cells (ADSCs) with epicardial deposit of ADSC sheets in a rat model of chronic myocardial infarction [69]. The aim of the study was to overcome some shortcomings related to intramyocardial injection, such as tissue damage and genesis of clusters of cells that eventually act as electrical barriers leading to malignant arrhythmias. Firstly, ADSC sheets exhibited superior cell engraftment over the ADSC-injected group. Moreover, CD90 and enhanced green fluorescent protein (eGFP) immunostaining revealed the ability of these cells to migrate in the ADSC sheets group. In contrast, few cells were recognized among cardiomyocytes in the ADSC-injected group. Secondly, a significant remodeling of left ventricle (LV) was shown in rats receiving intramyocardial ADSC (*p* = 0.05) in contrast to the cell sheet group. Finally, higher survival was documented in the cell sheet group (*p* = 0.001) [69]. Merging cell sheets with EPCs might be a key strategy to improve exponentially the performance of the sole approaches. Kobayashi and colleagues seeded green fluorescent protein (GFP)-positive EPCs on fibroblast sheets in order to obtain co-cultured cell sheets [70]. They compared these sandwich-like constructs with fibroblast sheets, EPC injection, and controls in infarcted hearts. Heart ultrasound was performed at baseline and weekly scheduled for 4 weeks. With respect to functional recovery, they showed a recovery trend of fractional shortening (FS) in the fibroblast sheet group and EPC group, but echocardiography revealed a significant improvement in only the co-cultured cell sheet group compared with the others. In addition, they assessed the ratio of connective tissue in the infarcted area with Azan stain and documented a significant connective tissue shrinkage in the co-cultured cell sheet group compared with controls and sole EPC group (*p* < 0.05). In light of these findings, they advanced that the exploitation of combined approach might create a kind of synergy: fibroblast sheets may have stimulatory effects on host endothelial cells and they may induce the proliferation and differentiation of the inserted EPCs. Notably, harnessing the capacity of subcutaneous granulocyte-macrophage colony-stimulating factor (G-CSF) to upregulate EPCs, combined with intramyocardial stromal cell-derived factor 1a (SDF-1a)-mediated EPC chemokinesis, showed success in the preservation of myocardial function in murine models of AMI. Interestingly, a separated employment of either G-CSF or SDF did not provide a similar finding [71]. Cardiac stem cell (CSC) sheets combined with concomitant EPC transplantation provided interesting findings in a swine chronic ischemic injury model. At 8 weeks after treatment, the left ventricle ejection fraction (LVEF) of the CSC–EPC combined group was demonstrated to be significantly higher compared with the EPC only group (*p* < 0.001) [72]. Moreover, the combined approach correlated with less accumulation of collagen in the area of chronic ischemic injury rather than separated approaches (*p* < 0.001) [72]. These findings could be ascribed to the ability of EPCs to promote host tissue expression of angiogenic cytokines, such as SDF-1, leading to an improvement in the migration of the transplanted CSCs. In consideration of miR-126-3p capability to modulate angiogenesis by regulating pro-angiogenic cytokine expression [73,74], Li et al. combined an EPC-based approach with gene therapy in an ischemic cardiomyopathy (ICM) murine model [75]. Basically, they collected EPCs from patients with ICM and transplanted them in nude rats, after miR-126-3p overexpression by recombinant lentiviral transfection. miR-126-3p-overexpressing EPCs (MO-EPCs) were demonstrated to be linked to a significant reduction of infarct size (*p* = 0.032) compared with both control and blank vector groups. They documented MO-EPCs to be significantly linked to an increase in capillary density (*p* = 0.022) as well as an improvement of LVEF (*p* < 0.05). Interestingly, the cytokine array revealed the upregulation of some cytokines such as G-CSF, VEGF-A, angiogenin, and hepatocyte growth factor (HGF) and the downregulation of others such as IL-8, tumor necrosis factor (TNF)-α, and TNF-β in the MO-EPC group. Nevertheless, no difference in the survival time of nude rats was found during 8 weeks of observation [76]. To sum up, although clinical trials project EPCs in clinical practice, transplant approach remains a crucial subject of debate. EPCs seem to play a crucial role in CVDs, but we need to clarify the best strategy of their employment.

## 6. Endothelial Progenitor Cells (EPCs) in Regenerative Medicine in Humans for Cardiovascular Disorders: State of the Art

Cardiovascular disease (CVD) represents notoriously the leading cause of death worldwide. This scenario is expected to remain unchanged over the years, despite various strategies for diagnosis and treatment: deaths for CVD are projected to be about 22 million in 2030, most of them attributable to coronary artery disease (CAD) [77]. An association between systemic inflammation and endothelial dysfunction in a wide spectrum of diseases has been consistently demonstrated [32,78,79,80,81] as well as the embryonic relationship between endothelium and EPCs [10]. Although a drop in the number of circulating EPCs was found in subjects with impaired vascular function, [82,83], a 5.8-fold increment of total circulating CD34^+^ cells (*p* < 0.001) was documented in high cardiovascular risk patients AMI [84]. Thanks to their landmark work, Asahara and co-workers disclosed the potential of EPCs in postnatal neovascularization by showing the recruitment of EPCs into foci of neovascularization in murine myocardial ischemia model [5,10]. More specifically, both “early” and “late” EPCs are considered to be crucial, albeit in a different manner. Early EPCs are thought to act in a paracrine manner by releasing key pro-angiogenic factors, such as VEGF and SDF-1. They promote the recruitment of “late” EPCs, facilitate their incorporation into new capillaries, and finally, enable them to rebuild the vascular network [11,18,85]. Since 1997, various studies have extensively investigated the achievable translation of these findings in clinical practice. Steinhoff et al. combined coronary artery bypass grafting (CABG) revascularization to autologous CD133^+^ bone marrow stem cell (BMSC) transplantation into the infarction border zone in subjects with AMI. With regard to the left ventricular ejection fraction (LEVF) variance compared with baseline at 180 days, both transplanted patients and controls (CABG only) exhibited a significant improvement, but the difference between the groups was not statistically significant. The study also showed a reduction of scar size compared with controls (*p* = 0.023) in the absence of significant reduction of N-terminal pro-brain natriuretic peptide (NT-proBNP) value in CD133^+^ cell-treated patients. In a following post-hoc analysis, subjects were grouped into either responders (R) or non-responders (NR) according to an increase in LVEF of at least 5% at 180 days. A significant decrease in both end diastolic (*p* = 0.008) and end systolic (*p* = 0.0001) volume of left ventricular dimensions and reduction in NT-pro-BNP (*p* = 0.002) were found in responders in contrast to non-responders. The impaired angiogenic capacity in the NR group was supposed to be related to elevated SH2B3 gene expression, although further clinical evaluations were required (PERFECT trial) [86]. Turan et al. published similar findings in 38 patients. In their trial, CD34^+^ and CD133^+^ bone marrow cells (BMCs) were isolated from bone marrow aspiration and intracoronarily injected by the stop-flow balloon catheter technique. They found a significant improvement of global ejection fraction in addition to a reduction of BNP levels and infarcted area at 3, 6, and 12 months after transplantation [87]. Comparable results were achieved in patients with anterior AMI. A significant increment of LVEF was detected after 4 and 12 months (*p* = 0.023 and *p* = 0.048, respectively) in patients treated with bone marrow-mesenchymal stem cells (BM-MSCs) compared with controls [88]. Intracoronary route administration was also evaluated in patients not eligible for either CABG or percutaneous coronary intervention (PCI). Clustered into two groups, patients were treated either with a low dose of CD34^+^ in group 1 or a high dose in group 2. To increase the number of circulating CD34^+^ cells, both groups received granulocyte colony-stimulating factor subcutaneously before leukapheresis, a less invasive collection procedure. In this study, Lee et al. documented a significant improvement in LVEF (evaluated using 3D-ecocardiography and magnetic resonance imaging (MRI)) and the severity of congestive heart failure (CHF) and angina pectoris. Intriguingly, they did not find any difference between the two groups with regard to clinical outcomes [89]. In their phase 2, randomized, double-blind, placebo-controlled study, Quyyumi et al. tested intracoronary administration of bone marrow-derived autologous CD34^+^ cells in ST-segment elevation myocardial infarction (STEMI) patients with reduced ventricular function. Although no association between LV end-systolic and end-diastolic volume and CD34^+^ cell dose appeared to subsist, authors found greater LVEF change in patients receiving >20 million CD34^+^ cells (10.2 ± 9.8%) compared with controls (*p* = 0.049). They also reported a decrease in major adverse cardiovascular events (MACE), although not statistically significant (*p* = 0.06) [90]. The absence of valvular disease, coronary artery disease, hypertension, and congenital heart disease in patients with structural and functional myocardial abnormality should point to the diagnosis of cardiomyopathy [91]. Vrtovec at al. investigated the role of CD34^+^ transplantation in patients with dilated cardiomyopathy and the relationship between clinical response and stem cell homing. The study revealed a significant improvement with regard to LVEF, 6-min walk test distance, and NT-proBNP at 5 years. A divergence of clinical response was also noted between patients with good homing and those with poor homing. According to the authors, this contrast might be ascribable to advanced age and higher plasma IL-6 levels [92]. Moreover, CD34^+^ stem cells have been demonstrated to be capable of improving ventricular perfusion even in areas that are not directly treated. These findings suggest a global action rather than a local effect [93,94]. In conclusion, in the era of so-called tailored medicine, characterization of eligible patients as well as standardization of methods of administration, stem cells sources, and methods of collection are strictly required to assess both the safety and efficacy for treatment of CVDs and, finally, to assign the appropriate place to EPCs in the therapeutic armamentarium. 

## 7. Clinical Trials for Regenerative Medicine in Cardiovascular Ischemic Diseases Employing EPCs: Delivery Strategies

In the last fifteen years, there has been a growing interest about the potential role of stem cells in AMI and chronic ischemic heart failure (HF). In particular, EPC cells, because of their angiogenic capacity, were and still are the object of different studies [61]. Several randomized controlled trials (RCT) were conducted in this field (Table 1). One of the first studies to evaluate the feasibility and safety of intracoronary administration of CD133^+^ stem cells (SCs) in patients suffering from AMI, treated with stenting, was published in 2005. The conclusions of this study were not encouraging; indeed, the authors reported an increased incidence of coronary events in the group of patients treated with SCs [95]. Two years later, Manginas et al. assessed the safety and feasibility of the same technique in a single center study involving 24 patients affected by nonviable, old anterior myocardial infarction. They observed that intracoronary infusion of selected CD133^+^ and CD133^−^CD34^+^ progenitor cells was safe and could improve left ventricular function [96]. In patients with healed myocardial infarction, intracoronary infusion of EPCs was a safe procedure and a significant improvement in LVEF was demonstrated [97]. Data from 775 consecutive procedures of intracoronary administration of EPCs using the stop-flow technique to analyze periprocedural complications and 30-day outcome reported that the death, stroke, acute myocardial infraction, and heart failure rehospitalization rates were, respectively, 0.5%, 0.13%, 1%, and 0.64%, showing a similar trend with adverse effects of normal angiography study [98]. In another RCT, the CELLWAVE trial (NCT00326989), Assmus et al. demonstrated a mild increase in LVEF at 4 months after intracoronary administration of BMCs combined with targeted shock wave in patients suffering of postinfarction chronic heart failure [99]. Similar results for cardiac function were found in the multicenter, phase II, randomized, double-blind, and placebo-controlled trial, REGENERATE-AMI (NCT00765453). The investigators of the study assessed the effects of intracoronary delivery route of BMCs, performed within 1 day after successful primary percutaneous intervention (PPCI), on left ventricular function in patients with AMI and regional wall motion impairment. After 1 year, a small improvement in LVEF was observed using advanced cardiac imaging in the group of patients treated with BMCs compared with placebo group [100]. Moreover, Colombo et al. conducted an RCT, NCT00400959, to assess the potential role of intracoronary CD133^+^ SC infusion on myocardial blood flow and function in patients suffering from acute STEMI. They found that intracoronary administration of bone marrow-derived, but not peripheral blood-derived, CD133^+^ SCs, performed within 10-14 days after STEMI, could improve long-term perfusion [101]. The safety of another SC delivery system, that used intramyocardial infusion of autologous CD133^+^ bone marrow SCs, during CABG was first evaluated in 2007 in an RCT including 40 patients. No procedure-related complications were reported in the 3 years of follow-up; myocardial perfusion and LVEF were both increased during the six-months of re-evaluation [102]. Another recent North American, multicenter, phase II, randomized study (IMPACT-CABG; IMPlantation of Autologous CD133^+^ in Patients Undergoing Coronary Artery Bypass Grafting), NCT01467232, reconsidered again the intramyocardial BMC delivery route during CABG. Early results of the trial confirmed the safety and feasibility of this kind of treatment and no major adverse cardiac events were detected in the six months of follow-up. Three-year follow-up data in patients with severe CAD and refractory angina, after intramyocardial injection of autologous bone-marrow derived mesenchymal stromal cells, demonstrated no short- or long-term side effects and reduced hospital admission rates for cardiovascular disease, indicating a plausible impact on disease progression [103]. The HEALING-FIM (Healthy Endothelial Accelerated Lining Inhibits Neointimal Growth-First In Man) study reported that, in patients with de novo coronary artery disease, implantation of EPC-capture stents is a safe and feasible procedure [104]. However, data from another study reported a higher angiographic late loss in STEMI patients [105]. A combination of paclitaxel-coated balloons with EPC-capture stents revealed that re-stenosis rate was reduced from 23.2% to 5.1% and no evident stent thrombosis was observed during the three months of follow-up [106]. After the 5-year follow-up, this combination demonstrated a lower clinically driven target lesion revascularization rate and less major adverse cardiac events [107]. Even though the above-mentioned data regarding short-term follow-up were promising, long-term follow-up failed to show efficacy and increased risk for restenosis was registered [108]. The Harmonized Assessment by Randomized Multicenter Study of OrbusNEich’s Combo StEnt (HARMONEE) trial reported that the combination of sirolimus and an abluminal bioabsorbable polymer with a novel endoluminal anti-CD34^+^ antibody coating designed to capture endothelial progenitor EPCs was not inferior to the everolimus-eluting stent treatment strategy [109].

The IMPACT-CABG II trial will estimate eventual improvement in perfusion and myocardial function [110]. In conclusion, as suggested by the European Society of Cardiology more than ten years ago, further RCTs are needed to assess long-term effectiveness and clear indications for the use of stem cell therapy in myocardial ischemia; nevertheless, the safety and feasibility of this treatment have been confirmed in several different RCTs [111]. 

## 8. Limitations and Criticisms of Clinical Trials in the Field

EPCs generated from bone marrow or peripheral blood have been applied in clinical studies as discussed above. However, there are several critical factors that determine the success of the regenerative capacity of EPCs and make their use difficult. As a result, despite the expectations, the speed of translation of clinical trials into clinical practice is quite slow.

The first consideration is that the ability to successfully isolate ECFCs is crucial prior to its consideration for clinical use [112]. Indeed, there is a wide heterogeneity in isolated cell types in many clinical trials, making difficult the comparison of results among studies and the understanding of functional cell types [112]. The samples and tissues of derivation of EPCs are also heterogenous (blood and bone marrow) and a clear guideline for cells and samples is not available. 

Secondly, there is a lack of standardization of cell surface markers and culture condition protocols when producing ECFCs for therapeutic intervention. Although Medina et al. highlighted the importance of EPC identification with more than one surface marker and information about isolation and culture conditions, in many clinical trials, surface markers for used cells are not specified [11]. 

Several data suggest that systemic inflammation compromises the reparative properties of endothelial progenitor cells [113]. Indeed, prolonged and unresolved inflammation post-ischemic heart event significantly compromises EPC reparative function in myocardial repair [114,115]. Therefore, pharmacological treatments are often required to achieve good regenerative results. Among others, IL-10 supplementation seems to be crucial [113]; on the other hand, IL-10 deficiency impairs EPC survival and function in ischemic myocardium [116]. Encouraging results support the use of statins in regenerative therapy to augment the number and function of EPCs in vivo to repair damaged tissues [117]. Recent data from the PROCREATION (PROgenitor Cells Role in Restenosis and Progression of Coronary ATherosclerosis After Percutaneous Coronary Intervention) study reported that increased numbers of circulating CD34^+^/KDR^+^/CD45^−^ cells is a predictor of major cardiovascular adverse outcome during long-term follow-up [118].

Besides the difficulties in isolation and culture, the major limitations of EPC therapy are the long culture times to generate a therapeutic dose [112] and that it can only be administered in an autologous fashion due to its inherent immunogenicity [112]. One of the major obstacles associated with stem cell survival in regenerative medicine is immune rejection [119]. However, the use of immunosuppressants might help in reducing rejection, although this solution is difficult to be realized. Recently, the sustained release of an immunosuppressant with the purpose of improving the survival of stem cells was successfully realized by nanoparticle-anchoring hydrogel scaffolds that also showed low toxicity [120]. On the other hand, scaffolds (natural biomaterials such as collagen, chitosan, and cellulose; and synthetic biomaterials such as polycaprolactone (PCL), polylactic acid, polyethylene glycol, and polyvinyl pyrrolidone) provide a suitable 3D niche, helping the cells to grow, proliferate, and differentiate [121].

Finally, other factors that need to be considered are cell dosage, delivery route, and timing of EPC administration [122].

In conclusion, additional clinical data are required to conclusively define the use of EPCs in the clinical practice for regenerative medicine.

## Figures and Tables

**Figure 1 cells-09-01886-f001:**
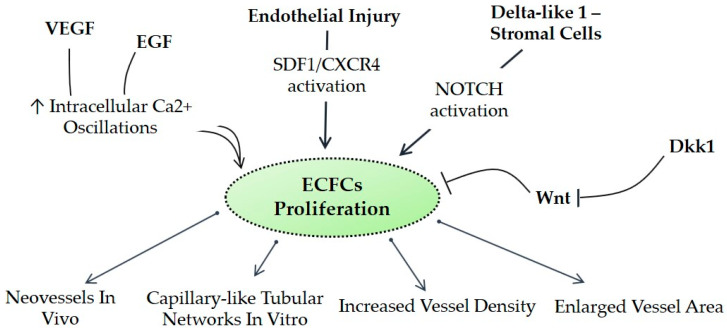
Postulated signaling pathways affecting endothelial progenitor cells (EPCs). CXCR4, C-X-C chemokine receptor type 4; Dkk1, dickkopf WNT signaling pathway inhibitor 1; ECFCs, endothelial colony-forming cells; EGF, epidermal growth factor; SDF-1, stromal cell-derived factor 1; VEGF, vascular endothelial growth factor; WNT, wingless-type mouse mammary tumor virus integration site family.

**Table 1 cells-09-01886-t001:** Clinical trials for regenerative medicine in cardiovascular ischemic diseases employing EPCs.

Cell Type	Number of Patients Treated (Controls)	Safety Feasibility	Technique	Findings	Limitations	References
CD133^+^ BMSCs	19 (16)	+	Intracoronary injection	↑ ejection fraction * ↑ LVSP/LVESVI ↓ chordae shortening * ↓ MIBI perfusion defect *	↑ incidence of coronary events at 4-month follow-up	[95]
CD133^+^ CD34^+^133^−^ BMSCs	12 (12)	+	Intracoronary injection	↑ ejection fraction ^#^ ↑ myocardial perfusion ↓ ED and ES volume ↓ ventricular Remodelling - No apparent major adverse cardiac events	- Lack of a randomization group - Absence of PET and MRI evaluation - Short follow-up	[96]
BMSCs	42 (60)	+	Intracoronary injection + shock wave	↑ LV ejection fraction ↓ NYHA class ↓ NT-proBNP	- Advanced heart failure - Low numbers of administered BMSCs - Absence of MRI evaluation for some patients	[99]
CD133 selected/CD34^+^ BMSCs	20 (20)	+	Intramyocardial delivery	↑ LV systolic function	- Absence of MRI evaluation - Heterogeneity of the preoperative LV contractility - No LV volume detection - Small number of BMSCs	[102]
CD133^+^, CD34^+^, CD45^+^ BMSCs	19 (14)	+	Intramyocardial injection	↑ LV systolic function	- Small number of patients	[110]

BMSCs, bone marrow stem cells; ED, end diastolic; ES, end systolic; LV, Left Ventricle; LVSP, peak left ventricular systolic pressure; LVESVI, left ventricular end-systolic volume index; MIBI, Tc 99m sestamethoxyisobutylisonitrile; MRI, magnetic resonance imaging; NT-proBNP, N-terminal pro-brain natriuretic peptide; NYHA, New York Heart Association; PET, positron emission tomography; ↑, increased; ↓ decreased * *p* < 0.05; ^#^
*p* = 0.016.

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
