# Peer review of "The State of Art of Regenerative Therapy in Cardiovascular Ischemic Disease: Biology, Signaling Pathways, and Epigenetics of Endothelial Progenitor Cells"

_cells, 2020, doi:10.3390/cells9081886_

Round 1
Reviewer 1 Report
The review aims at providing an overview of regenerative therapy, calcium modulation and the molecular pathways involving NOTCH, VEGFR, CXCR4 that contribute to endothelial progenitor cells (EPCs) proliferation and differentiation. There has been significant progress of EPCs biology and regulatory pathways that may contribute to applications in the clinical practice it an important topic to review.
The following comments emerged from reading this interesting Review:
Major:
- The title does not fully contemplate what is mentioned in the text, giving the false impression that it will be a review about intracellular calcium modulation in EPCs biology, instead, it is mentioned in one paragraph related to signaling pathways. I would suggest “The state of art of the regenerative therapy in cardiovascular ischemic disease: biology, signaling pathways and epigenetics of endothelial progenitor cells.” or similar.
- There are no figures related do the “signaling pathways driven by or affecting EPCs” with only the text it is hard to follow and understand.
- In the clinical trial, a table showing the main clinical trial findings would complement the text. Please, provide the limitations and the authors criticism of the clinical trials in the field.
Minor:
- Ca2+ should 2+ be superscript;
- In vivo and in vitro should be italics;
- Ca+1-chelant should be a Ca+2-chelant: EDTA or EGTA?
Author Response
Reviewer 1
The review aims at providing an overview of regenerative therapy, calcium modulation and the molecular pathways involving NOTCH, VEGFR, CXCR4 that contribute to endothelial progenitor cells (EPCs) proliferation and differentiation. There has been significant progress of EPCs biology and regulatory pathways that may contribute to applications in the clinical practice it an important topic to review.
The following comments emerged from reading this interesting Review:
Major:
- The title does not fully contemplate what is mentioned in the text, giving the false impression that it will be a review about intracellular calcium modulation in EPCs biology, instead, it is mentioned in one paragraph related to signaling pathways. I would suggest “The state of art of the regenerative therapy in cardiovascular ischemic disease: biology, signaling pathways and epigenetics of endothelial progenitor cells.” or similar.
We thank the referee for this comment. We agree that the title did not reflect the aim of this research and we have accordingly modified the title.
- There are no figures related do the “signaling pathways driven by or affecting EPCs” with only the text it is hard to follow and understand.
We thank the referee for this important criticism arisen; we have now included a figure summarising the complex interaction between ECPs and related pathways
- In the clinical trial, a table showing the main clinical trial findings would complement the text. Please, provide the limitations and the authors criticism of the clinical trials in the field.
We thank the referee. We have included a table with relevant findings of the reported clinical trials. We also thank the referee for highlighting the importance of describing the limitations of this field of research. We have therefore included a novel paragraph for depicting the barriers that limit this approach the from bench to bedside.
Minor:
- Ca2+ should 2+ be superscript;
We have modified.
- In vivo and in vitro should be italics;
We have modified.
- Ca+1-chelant should be a Ca+2-chelant: EDTA or EGTA?
We have modified
Reviewer 2 Report
The authors wrote a review focusing on the role of endothelial progenitor cells (EPCs) for myocardial regeneration. Overall, the main theme of this review has been addressed repeatedly in the existing literature (ex. Pharmacol Ther 2018;181:156-68, Br Med Bull 2017;121:135-54, Stem Cells Int 2016;2016:8340257, and many more). Many aspects of the potential mechanisms involved in the experimental benefits conferred by EPCs have been reviewed and summarized before, including exosomal communication, neovascularization, plasticity alteration, etc. From this perspective, the theme of this review appears relatively obsolete, as this topic is now considered controversial and little new information is available during the recent 2-3 years. Moreover, the references they cited to support their descriptions in the benefit of EPCs in myocardial regeneration are rather outdated (the most recent one related to clinical and experimental parts are from 2017~2018 [NO. 66~94; NO.80 is an unrelated one]). Since the most recent review of the same topic is also 2 years ago (2018) and the supporting information of this review does not extend beyond their understandings, it is difficult to assign sufficient priority for this review. This is the main defect of this review. Other issues are briefly listed below.
- In the review, the authors initially addressed EPCs as a whole and then turned to endothelial colony forming cells (ECFCs) for mechanistic descriptions, which is rather confusing. Multiple types of EPCs have been examined before with their potential mechanisms proposed or tested, but only ECFCs are mentioned regarding mechanistic discussions. Enrichment is definitely needed.
- More than two-thirds of this review cover background information and technical details of EPCs unrelated to myocardial regeneration and relevant issues. This also weakens their claim and lowers the novelty of this review.
Author Response
Reviewer 2
The authors wrote a review focusing on the role of endothelial progenitor cells (EPCs) for myocardial regeneration. Overall, the main theme of this review has been addressed repeatedly in the existing literature (ex. Pharmacol Ther 2018;181:156-68, Br Med Bull 2017;121:135-54, Stem Cells Int 2016;2016:8340257, and many more). Many aspects of the potential mechanisms involved in the experimental benefits conferred by EPCs have been reviewed and summarized before, including exosomal communication, neovascularization, plasticity alteration, etc. From this perspective, the theme of this review appears relatively obsolete, as this topic is now considered controversial and little new information is available during the recent 2-3 years. Moreover, the references they cited to support their descriptions in the benefit of EPCs in myocardial regeneration are rather outdated (the most recent one related to clinical and experimental parts are from 2017~2018 [NO. 66~94; NO.80 is an unrelated one]). Since the most recent review of the same topic is also 2 years ago (2018) and the supporting information of this review does not extend beyond their understandings, it is difficult to assign sufficient priority for this review. This is the main defect of this review. Other issues are briefly listed below.
- In the review, the authors initially addressed EPCs as a whole and then turned to endothelial colony forming cells (ECFCs) for mechanistic descriptions, which is rather confusing. Multiple types of EPCs have been examined before with their potential mechanisms proposed or tested, but only ECFCs are mentioned regarding mechanistic discussions. Enrichment is definitely needed.
- More than two-thirds of this review cover background information and technical details of EPCs unrelated to myocardial regeneration and relevant issues. This also weakens their claim and lowers the novelty of this review.
We wrote this review taking into account the presence of other previous reviews, trying to report the latest updates, considering the topic very current given the growing number of ischemic events in the world. The bibliographical references present are the most up to date and the presence of a limited number of new studies shows that the topic is very complex.
In the introductory part of the review we have tried to provide a classification of EPCs, according to the most recent classification used; surely, as also said in the manuscript (“there is a controversy between correct theoretical definition and scientific evidence for the identification and definition of the role of these cells, causing the denomination of numerous cell types as EPCs”) confusion arises, as already said by Prasain, N. et al. 2012 and Medina, R.J.et al. 2011; they suggest to replace the term EPCs with ECFCs, precisely because of their phenotype and their biological functions, well describing these cell types. Precisely for this reason we focused our attention in particular on the description of the ECFCs, cells which, among all, have the greatest ability to generate mature endothelial cells; differently the MACs do not give rise to endothelial cells, but remain faithful to their hematopoietic nature (Kanayasu-Toyoda, T.et al. 2016; Urbich, C., et al. 2003).
Therefore, in the revised manuscript we have made a major revision which includes:
1- a figure summarising the complex network between the EPCs and different signalling pathways
2- a table for better dissect the relevant findings of early researches using EPCs in clinical models
3- a novel paragraph in which the main the barriers limiting this approach the from bench to bedside.
We hope you may consider the efforts we made in improving the overall quality of the manuscript.
Reviewer 3 Report
This is an interesting review of the potential benefits of regenerative therapy in cardiovascular medicine.
Authors summarize experimental and clinical data and suggest possible clinical applications.
Reading the manuscript looks rather demanding for the average Cardiologist that would appreciate a list of abbreviation.
A useful addition would be a paragraph to discuss why an hypothesis that has been around for decades, is still far from clinical use.
Author Response
This is an interesting review of the potential benefits of regenerative therapy in cardiovascular medicine.
Authors summarize experimental and clinical data and suggest possible clinical applications.
1- Reading the manuscript looks rather demanding for the average Cardiologist that would appreciate a list of abbreviation.
We have now included an abbreviations list.
2- A useful addition would be a paragraph to discuss why a hypothesis that has been around for decades, is still far from clinical use.
We thank the referee. We have therefore included a novel paragraph depicting the barriers that limit this approach the from bench to bedside.
Round 2
Reviewer 2 Report
The authors have made some improvements to their manuscript content. I cling to the original stance that the idea is rather obsolete. However, the English style is quite suboptimal and really needs English polishing.